# 🦀CRAB: Assessing the Strength of Causal Relationships Between Real-World Events

**Angelika Romanou,  Syrielle Montariol,[*]  Debjit Paul,[*]**
**Léo Laugier,  Karl Aberer,  Antoine Bosselut**
EPFL
{firstname.lastname}@epfl.ch

## Abstract

Understanding narratives requires reasoning about the cause-and-effect relationships between events mentioned in the text. While existing foundation models yield impressive results in many NLP tasks requiring reasoning, it is unclear whether they understand the complexity of the underlying network of *causal relationships* of events in narratives. In this work, we present *CRAB*, a new **C**ausal **R**easoning **A**ssessment **B**enchmark designed to evaluate causal understanding of events in real-world narratives. *CRAB* contains fine-grained, contextual causality annotations for $\sim$ 2.7K pairs of real-world events that describe various newsworthy event timelines (*e.g.*, the acquisition of Twitter by Elon Musk). Using *CRAB*, we measure the performance of several large language models, demonstrating that most systems achieve poor performance on the task. Motivated by classical causal principles, we also analyze the causal structures of groups of events in *CRAB*, and find that models perform worse on causal reasoning when events are derived from complex causal structures compared to simple linear causal chains. We make our dataset and code available to the research community.[1]

## 1 Introduction

Understanding narratives requires understanding the cause-and-effect relationships between interconnected sub-events of those narratives. When reading text, humans immediately induce potential causal links between the events presented as part of a larger scenario (Grunbaum, 1952; Pearl and Mackenzie, 2018). For example, in Figure 1, when reading an article about the acquisition of Twitter in 2022, a reader would implicitly assign causal links between events such as E2: "Elon Musk closes 44 billion dollar deal to buy Twitter" and E3: "Twitter delists from the NYSE".

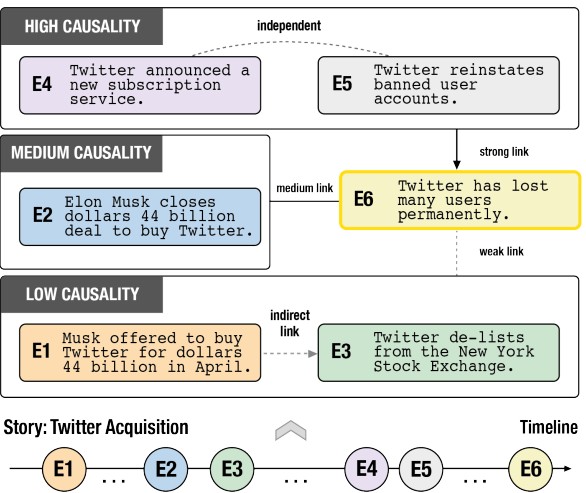

Figure 1: Events from *CRAB* that lead to event E6, forming causal sub-structures with links of various causal strength.

However, building accurate causal mental models of the situations depicted in narratives poses several complex challenges. First, human causality judgments are rarely binary. Instead, they fall on a spectrum depending on human perception of other mediating or confounding events (Pearl, 2009). For example, in Figure 1, E2 is a mediator event for the causal relationship of E1 and E3, likely affecting the human perception of the causal relationship between E1 and E3. Second, causality judgments depend on the context depicting the events in question — a context that can affect perceptions of causality. For example, a high causal judgment might be assigned between E4 and E6 in Figure 1. However, the introduction of new information about E5, another potential cause of E6, might downgrade the perceived intensity of a causal link between E4 and E6. Finally, because context is critically important to judging causal relationships about events, and most narratives offer an incomplete (and sometimes biased) reporting of particular scenarios, multiple sources may be required to paint an accurate picture of the causal relationships between multiple

---

[*]Equal contribution
[1]https://github.com/epfl-nlp/CRAB

interconnected events.

Addressing these challenges, we introduce *CRAB*, a new **C**ausal **R**easoning **A**ssessment **B**enchmark that contains fine-grained, contextual causality annotations of real-world events that happened in the past ten years and received extensive media coverage. To collect the proposed benchmark, we design a crowdsourcing framework motivated by standard causal principles from cognitive science (Cao et al., 2022) and actual causality (Halpern, 2016) that study how humans perceive and express causality and responsibility among events. Using this knowledge frame, we automatically extract the events of newsworthy stories by integrating large pre-trained LMs into the dataset creation loop. We then construct causal graphs — combinations of inter-connected events forming different causal chains and frames, as presented in Figure 2 — from the extracted events and assess the strength of the causal relationships between these events using human annotators.

Our resulting benchmark, *CRAB*, contains ~2.7K high-quality event pairs, their causal score, and the respective documents in which the events appeared. All the events are grouped into 1.8K causal frames and 352 causal chains. We use this benchmark to assess the abilities of state-of-the-art (SoTA) models to understand and reason over the causal relationships of real-world events present in a set of contexts (*i.e.*, online news articles). Our analysis reveals that LLMs can capture explicit causal statements through pre-training, but they face difficulty applying causal reasoning to new scenarios, limiting their generalization and accuracy in offering predictions and explanations. We further stratify our results based on the structures of causal frames and chains, showing that they struggle with assessing the causality between events derived from complex causal structures compared to simple linear causal chains, especially when these events are extracted from different documents.

## 2 Preliminaries on Causality

In this section, we define the main causality concepts that we use to create and analyze *CRAB*.

**Actual Causality**    Actual causality refers to the causal relationship between specific events and their causes in the real world (Halpern, 2016) and seeks to understand the precise mechanisms by which one event leads to another, going beyond mere correlation. Understanding actual causality is crucial for humans to comprehend the underlying factors driving specific events, helping them make sense of the world, predict outcomes, and take practical actions toward those outcomes. Research in causal inference has attempted to formalize actual causality using causal models that map how humans perceive and attribute cause and responsibility to events and their outcomes. However, human perception of causality usually depends on background context, implicit biases, epistemic state, and lack of information, making the task of actual causality attribution challenging to formalize (Matute et al., 2015; Henne et al., 2021). Additionally, in cases where the responsibility of an event can be attributed to more than one preceding event, observers tend to assign different attribution to the contributing causes (Wolff and Shepard, 2013). Therefore, when events are described with natural language, the causal judgments are not binary but relative, enabling comparisons between causal events (Icard et al., 2017).

**Causal Frame**    Humans tend to attribute different degrees of causality between contributory events, relying primarily on domain and commonsense knowledge (Kıcıman et al., 2023). Causality research refers to this set of candidate events that are relevant to a particular outcome event as a Causal Frame (Halpern, 2016). An example of this can be seen in Figure 1, where the causal frame of event E6 comprises events E2, E4 and E5. We construct *CRAB* to collect causal frame subgraphs, where each event is associated with its potential causes, along with a causal score labeled by humans. In that way, a subset of events can be directly compared using their scores, a proxy for their likelihood to be the cause of a target event. Later in this paper, we explore the different types of Causal Frames extracted from *CRAB* as presented in Figure 2 (left) and we stratify our analysis on LLM assessment based on these structures.

**Causal Chain**    Another perspective for assessing the degree of causality between two events is to explore the chain of events that happened through time and led to an outcome event E (Pearl, 2009). We define the causal chain of an outcome event as the set of paths ending at E in the event's causal graph (see Section 3.1). In the example in Figure 1, the following causal chain can be considered: event E1 led to event E2, which led to event E3. In contrast with causal frames, the concept of the

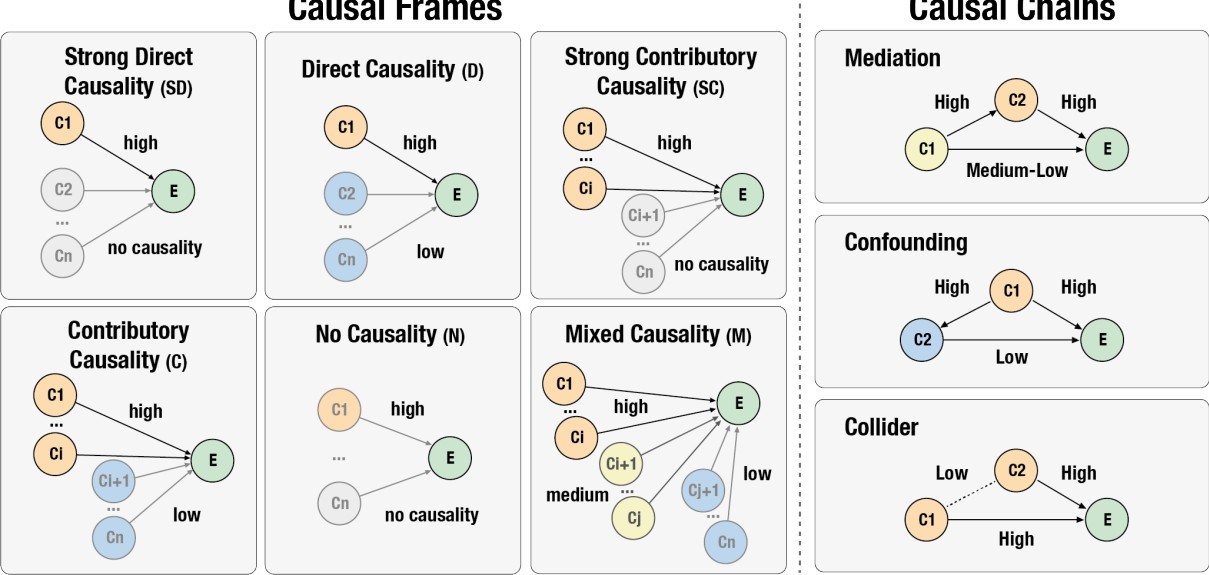

Figure 2: Different structures of causal frames (left; Halpern, 2016) and causal chains (right; Pearl, 2009) present in **CRAB**. The patterns in structures are formulated based on the different causal judgment scores among events. The colors of cause-nodes represent the causality strength they have towards the event-node E.

causal chain is heavily related to temporality since the events present in the chain are ordered not only based on causality but also by the time they took place. **CRAB** leverages both concepts of causal chains and temporality, providing a testbed for language models to assess their ability to perform causal reasoning in different causal chain structures as depicted in Figure 2 (right).

## 3 Dataset Construction

In this section, we give an overview of causal event linking and its associated challenges, and describe our approach for building our **CRAB** benchmark centered around these challenges. We create a corpus of documents from which we automatically extract events, build a timeline, and collect causality judgments between event pairs. The full data creation pipeline is described in Figure 3.

### 3.1 Overview

We consider a set of documents covering a news story. Each document reports several events associated with that story. The time-ordered list of these extracted events defines a *timeline* associated with the story. From a timeline and its set of documents, we can build a *causal event graph*. Our goal is to identify the causal relations between the events in the graph using the documents in which these events are mentioned.

**Challenges** Real-world causality identification and assessment poses several challenges:

*Subjectivity:* Causal judgments may be affected by the implicit knowledge and bias of the reader, rather than the content of the document. Readers may feel a strong impression of causation about an event, even when the real causal relation is unclear based on the context (Wolff and Shepard, 2013).

*Contextualization:* Limited or subjective coverage of stories by the documents makes the automated creation of the story timelines and the respective causal graphs subject to incompleteness.

*Temporality:* Extracting events from a single document and ordering them based on their occurrence in time is a relatively straightforward task. However, this task is more complex in a multi-document setting as extracted events are grounded in different documents, which may be published on different dates. Consequently, their respective timelines may be interleaved.

While methods have been proposed for causal link identification and assessment, they mostly focus on binary commonsense causality detection across short, single-document text (Zhang et al., 2023), which does not adequately address the unique challenges of real-world causality judgment. In the following sections, we present our approach for building **CRAB** that addresses the above challenges.

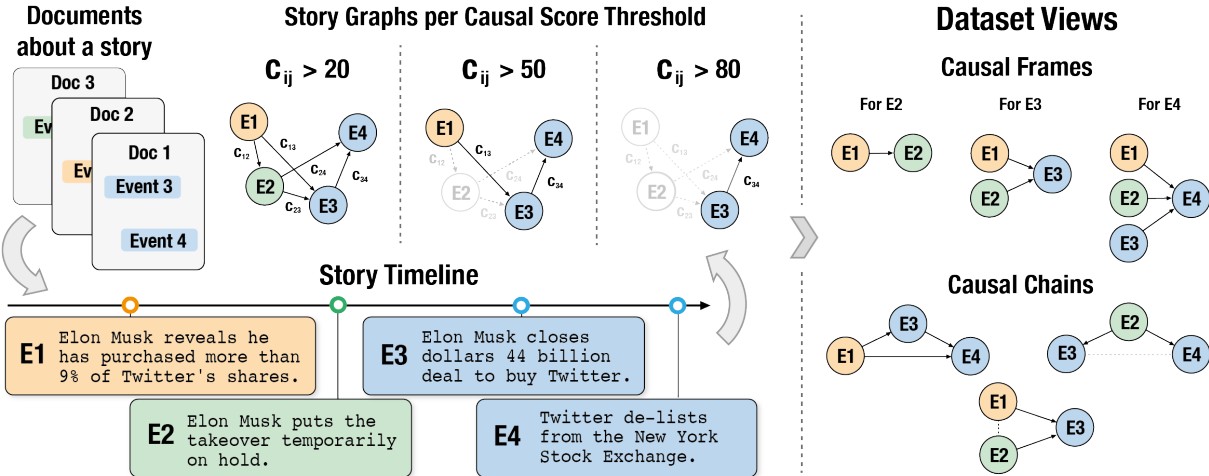

Figure 3: *CRAB* data pipeline overview: We collect documents covering newsworthy stories, create a timeline with the main events extracted from the documents for each story, and crowdsource human annotations on the causal judgments between the events *(score 0 to 100)*. Based on these scores, we generate a causal graph for each story that can be filtered on different causal score thresholds. *CRAB* can also be viewed from the perspective of causal frames and causal chains as presented in Section 2. Same-color events originate from the same document.

## 3.2 Document Selection

We first select 20 stories about major events that happened around the world during the past 10 years, covering geopolitical, social, and environmental themes. We scrape the top 20 news articles per story from Google News API,[2] filter the invalid ones and clip them to 250 words.[3] Our final corpus contains 173 documents, which we use for event extraction (see descriptive statistics and selection process details in Appendix A.1 and Table 17).

## 3.3 Event Extraction

We extract the main events mentioned in each document. In contrast to prior work (Shi and Lin, 2019), we use a generative approach to extract the main events given a news piece. Specifically, we prompt GPT-3 (text-davinci-003, Ouyang et al., 2022) to extract the main events from a given document (similar to Veyseh et al., 2021, see Appendix Table 12 for exact prompts). Extensive experimental analysis (see Table 10) confirms that this generative approach for event extraction leads to higher extraction precision and semantic granularity suited for our task.

**Extraction Verification** Because generative methods come with the limitation of hallucinations and wrong outputs, we filter extracted events to keep only the valid generations. We first automatically filter events that contain less than 3 tokens,

[2]https://serpapi.com/
[3]When clipping articles, we avoid dividing sentences.

and remove generations that represent claims or beliefs by filtering the utterances with conversational verbs followed by quotes (e.g. "*said*" + "*«*"). Then, two experts manually evaluate the validity of all the remaining generated events. After filtering, our dataset contains 384 unique events.

**Timeline Construction & Event Disambiguation** As causality is conditioned on temporality, we manually create a timeline of the extracted events for each of the 20 stories, by considering all documents associated with the story. While building these timelines, we disambiguate the events mentioned in different documents by merging differently phrased instances of the same event. Post-annotation, we refer to the merged instances as one single node in the causal graph while keeping all the textual versions of it in the dataset.

## 3.4 Event Causality Linking

In the final stage of our pipeline, we collect causality judgments about all event pairs extracted from the documents related to a specific story (2730 pairs). We distinguish two settings: when both events in the pair are originally from the same document (*in-document* setting, 360 pairs) and when they originate from different documents (*cross-document* setting, 2370 pairs). Motivated by the way cognitive studies capture judgments about actual causality (Gerstenberg et al., 2021), we define the causality between real-world events not as a single binary score but as a continuous value from

0 to 100, enabling finer analysis and predictions of causal judgment. We qualify 44 Amazon Mechanical Turk workers, and for each pair of events, task 7 workers with providing a judgment for the causal link between the events (see Appendix A.3 for details).

### 3.5 Causality Strength Validation

We divide the causality score into 4 classes (0-20, 20-50, 50-80, 80-100) to compute Krippendorff's $\alpha$[4] between our 44 annotators (see Appendix A.4). Krippendorff's $\alpha$ for all classes is 0.28, and 0.38 when considering only the furthest classes (0-20 and 80-100).[5] We average all annotators' causality scores for each event pair to obtain a unique scalar causal judgment and classify this event pair score into the 4 causal classes (high, medium, low, and no causality). Given the low agreement, we select pairs where the average score falls on the boundary of the 4 classes and the variance between annotators is high, to be validated by experts. This subset consists of 26.7% of the benchmark. This step is done by asking three expert annotators (NLP researchers who are familiar with the task of causal inference) to further annotate event pairs' causal scores and classes. The experts, given the average causal score, were asked to choose which of the neighboring classes was a better class for the event pair, updating the score accordingly. The inter-rater agreement, using Krippendorph's alpha, between experts is 0.70. These expert-validated causal scores along with the remaining low-variance samples are used for **CRAB**.

## 4 Dataset Analysis

### 4.1 Descriptive statistics

**CRAB** consists of a set of 173 documents regarding 20 different stories discussing newsworthy real-world events. It contains 384 extracted unique event instances and 2730 event pairwise causality scores (see Table 5 in Appendix for additional descriptive statistics). The experiments presented in

---

[4]With https://pypi.org/project/krippendorff/, using interval metric to calculate the pairwise distance.

[5]We note that the high number of annotators per sample makes the raw number of disagreements higher, lowering $\alpha$. Moreover, contrary to classical annotation situations where a small number of annotators label each sample, annotation using Amazon Mechanical Turk involves many annotators participating in a task, thus each sample being annotated by different workers, augmenting the variance and decreasing the agreement rate.

| Type of pairs | Pairwise Event Causality Score | | | |
| --- | --- | --- | --- | --- |
| | **Below 20** | **20-50** | **50-80** | **Above 80** |
| **In-doc** | 3.9% | 25% | 26.6% | 44.4% |
| **Cross-doc** | 13.1% | 37.6% | 25.3% | 24% |
| **All pairs** | 11.9% | 35.9% | 25.5% | 26.7% |

Table 1: Percentage of pairs present in the **CRAB**, per causality score class.

the following section, are based on these 4 classes reported in Table 1.

### 4.2 Causal Structures

As discussed in Section 2 and based on expert annotations of the pairwise causality between events, we view **CRAB** from the perspective of causal frame subgraphs. We stratify the dataset and get the causal frame of each event. We categorize these subgraphs based on the causal links present in them. We observe different categories based on the strength of causal scores between the effect event and the potential causes (in-degree edges of the causal frame graph). Figure 2 (left) depicts the different types of causal frames present in the dataset, and Table 6 provides statistics relative to the number of chains and events for each one of the structure types.

Similarly to causal frames, we extract causal chains from **CRAB** based on the three causal structures; *Mediation*, *Confounding*, and *Collision* (Pearl, 2009), depicted in Figure 2 (right). Descriptive statistics of the extracted causal chains that fit into these three cases are reported in Table 7.

## 5 Experimental Setup

To evaluate how language models reason about causality, we define different experimental frameworks covering various causality assessment scenarios, similar to Kıcıman et al. (2023). We investigate three tasks in ascending order of complexity in terms of causal structure: pairwise causality inference, graded causality inference, and causal chain inference. This section describes the experimental setups associated with these tasks, and the models evaluated on them.

### 5.1 Models

We use two decoder-only instruction-following API-based models, *GPT-3* (text-davinci-003, 175B size) and *GPT-4*, with the settings suggested by OpenAI (2023): a *temperature* of 0.3 and a *max-*

*imum length* of 256 tokens. We additionally test **CRAB** using *Flan-Alpaca-GPT4-XL* (Chia et al., 2023), an open-source 3B size encoder-decoder model fine-tuned on instruction-following datasets: Flan (Longpre et al., 2023) and GPT4-Alpaca.[6]

## 5.2 Experiments

**Pairwise Causality Inference** To evaluate Pairwise Causality Inference, we first prompt the model to generate a scalar causality score between two events given a context (the documents from which the events were extracted), mimicking the benchmark human annotations. We compare it with the average annotators' score in **CRAB**. In initial experiments, this setup led to very poor performance as all the models failed to generate a scalar value for quantifying causality. Therefore, we mapped the causality intervals to descriptions of different degrees of causality and augmented the prompt instructions with score ranges and their explanations. The 4 classes and definitions are as follows (i) *High causality*: a link between events that are definitely causally related to each other (causal score above 80), (ii) *Medium causality*: a link between events that might be slightly causally related to each other (causal score between 50 and 80), (iii) *Low causality*: a link between events that have a little causal connection (causal score between 20 and 50), and (iv) *No causality*: independent events (causal score lower than 20). The full prompt can be found in Table 16 in Appendix. We then evaluate the model's answer by mapping the generated score to the four classes and computing the Macro-F1 score (**Pairwise Causality Score** in Table 2).

We also experiment with binary and multi-class classification tasks (**Binary Pairwise Causality** and **Multi-class Pairwise Causality** in Table 2, respectively), prompting the model to output a causality class instead of a raw score. For multi-class, the generated output is a letter matching one of 4 classes. For the binary classification, we ask the model whether an event caused another to happen based on the provided contexts, the answer being "yes" or "no". The prompts for these tasks can be found in Tables 13 and 14 in Appendix.

**Graded Causality Inference** As described in Section 2, one of the main concepts in actual causality is graded causation or responsibility (Halpern, 2016), which is the relative degree to which an

event causally contributes to an effect. For example, following the example in Figure 1, multiple events are responsible for causing E6: "Twitter has lost many users permanently", with different causality scores. Thus, we go beyond pairwise causality and design two methods to prompt models to rank the events that contributed more to the effect. First, we create a Multiple Choice Question (MCQ) Answering task that asks the model to provide the most contributory to an effect cause among several events. We construct the dataset for the experiment by using the causal frames of each event and selecting 4 possible causes. We then ask the model to select, based on these 4 choices, the cause with the highest causality score (see the prompt in Table 15 in Appendix).

Second, we assess the responsibility of event pairs by stratifying the results of the pairwise causality experiment based on the type of events' causal frames. For each event, we extract its causal frame (i.e., all the potential causes of an event) and predict the class of causality between that event and all the potential causes in the causal frame. We compute both the average F1-score and the Exact Match (EM) between **CRAB**'s causal class annotations (4 classes) and the causal class predictions across all event pairs in the causal frame. We report the scores in Table 3 and stratify them based on the different causal frame structures *strong direct (SD), direct (D), strong contributory (SC), contributory (C), mixed (M)* and *no causality (N)* (Figure 2 (left)).

**Causal Chain Inference** Pairwise causality provides a strong indication of the causal relationship between two events. However, these events are usually part of a larger chain of events with complex causal patterns. In this experiment, we consider not only the relations between the causes and the effect in causal frames but also how the causes are related to one another. We stratify the results of the pairwise causality experiment based on the three causal chain structures we introduced in Sections 2 and 4; *Mediation, Confounding,* and *Collider* (Figure 2 (right)). Similarly to the previous experiment, we extract causal chains that fit the three patterns and compute the F1-score and the Exact Match (EM) between **CRAB**'s causal class annotations (4 classes) and the causal class predictions for each edge in the causal chains. We report each model's average scores in Table 4.

---

[6] https://instruction-tuning-with-gpt-4.github.io/

| Tasks | Models | All pairs | In-doc | Cross-doc | Pre-Jan 2022 | Post-Jan 2022 |
|---|---|---|---|---|---|---|
| **Pairwise Causality Score** $C \xrightarrow{C_{ce}} E$ | Flan-Alpaca | 21.6 | 14.9 | 22.4 | 22.0 | 21.3 |
| | GPT-3 | 25.8 | 24.4 | 25.4 | 26.6 | 25.2 |
| | GPT-4 | **54.7** | **59.0** | **53.7** | **56.4** | **53.5** |
| **Multi-class Pairwise Causality** $C \xrightarrow[L/N]{H/M/} E$ | Flan-Alpaca | 11.0 | 12.2 | 10.8 | 11.2 | 10.7 |
| | GPT-3 | 35.0 | 27.4 | 34.9 | 35.0 | 34.5 |
| | GPT-4 | **45.6** | **46.1** | **45.0** | **43.1** | **46.7** |
| **Binary Pairwise Causality** $C \xrightarrow{0/1} E$ | Flan-Alpaca | 60.1 | 73.8 | 56.7 | 62.1 | 58.7 |
| | GPT-3 | 57.2 | 67.0 | 55.0 | 56.9 | 57.5 |
| | GPT-4 | **73.9** | **80.0** | **72.6** | **76.5** | **72.0** |
| **Graded Causality** *(MCQ)* | Flan-Alpaca | 39.9 | 53.2 | 29.3 | 44.1 | 35.4 |
| | GPT-3 | **59.7** | **70.9** | **50.7** | **64.5** | **54.5** |
| | GPT-4 | 53.8 | 67.3 | 43.1 | 63.3 | 44.5 |

Table 2: Macro F1-scores on SoTA LLMs on all *Pairwise Causality Inference* tasks as presented in Section 5.2 and the *Graded Causality Inference* MCQ task presented in Section 5.2. For the MCQ task, we stratify the results for in-doc and cross-doc based on whether the effect & correct cause are extracted from the same document.

| MODELS | All Frames | | SD | | D | | SC | | C | | M | | N | |
|---|---|---|---|---|---|---|---|---|---|---|---|---|---|---|
| | F1 | EM | F1 | EM | F1 | EM | F1 | EM | F1 | EM | F1 | EM | F1 | EM |
| Flan-Alpaca | 10.2 | 0.8 | 5.0 | 0.1 | 19.8 | 11.1 | 3.7 | 0.0 | 8.0 | 0.1 | 13.0 | 0.7 | 6.4 | 3.0 |
| GPT-3 | 28.8 | 3.3 | 29.1 | 4.1 | 30.5 | 16.6 | 31.9 | 3.3 | 31.0 | **4.1** | 29.0 | 0.0 | 19.3 | 9.1 |
| GPT-4 | **38.9** | **6.1** | 45.2 | 16.7 | 39.8 | 11.1 | 40.6 | 6.7 | 38.3 | 3.1 | 36.8 | 1.3 | 39.4 | 24.2 |

Table 3: Macro F1 and EM scores on SoTA LLMs for *Graded Causality Inference* (Section 5.2) stratified for different Causal Frame types. SD is for Strong Direct Causality; D is for Direct Causality; SC is for Strong Contributory Causality; C is for Contributory Causality; N is for No Causality; M is for Mixed Causality. Green colors depict the F1 and EM highest scores of the Causal Frame types for each LLM and orange colors depict the lowest scores. Please refer to Figure 2 for detailed visualization of the different *Causal Frame* structures.

| MODELS | Mediation | | Confounding | | Collider | |
|---|---|---|---|---|---|---|
| | F1 | EM | F1 | EM | F1 | EM |
| Flan-Alpaca | **49.4** | **25.4** | 38.1 | 10.0 | **29.3** | 8.2 |
| GPT-3 | 40.2 | 10.6 | 38.1 | 9.7 | 28.0 | **9.7** |
| GPT-4 | 38.2 | 5.8 | **44.9** | **20.9** | 25.1 | 5.4 |

Table 4: Macro F1 and EM scores on SoTA LLMs for the *Causal Chain Inference* stratified in different Causal Chain structures as described in Section 5.2. Green colors depict the F1 and EM highest scores of the Causal Chain types for each LLM and orange colors depict the lowest scores.

## 6 Experimental Results

In the previous section, we introduced several lower-level tasks to investigate whether various LLMs can accurately assess pairwise causality judgments. Here, we describe the results from the perspective of Causal Discovery, Causal Responsibility Assessment, and Multi-document Causal Reasoning. We perform an in-depth analysis of LLM capabilities in different facets of causality, grounded to principles and dimensions of actual causality described in Section 2.

**Causal Discovery** Table 2 provides the pairwise causality inference scores for the three pairwise sub-tasks. All models perform poorly on **CRAB**, with *GPT-4* showing a higher performance in most of the tasks compared to *GPT-3* and *Flan-Alpaca*. In the case of Binary Pairwise Causality inference, we measure *GPT-3* and *GPT-4*'s binary classification Macro-F1 scores for different thresholds (see Figure 5 in Appendix). Both models tend to predict causation between events with a gold causal score above *50*. Interestingly, if we increase this gold score threshold and consider only causal events with a high causal connection, the performance in all models drops. A possible interpretation behind these results is that large language models are typically quite good at identifying the distributional similarity between concepts. When it comes to making causal judgments, they may identify multiple related events as causally associated with an outcome rather than a single consequential event. This means that they may misclassify events weakly related to the event in question as causes. Ideally, when asked about a binarized causal relationship,

we would like models to provide only the main causes (strong relation between two events).

Models also under-perform when assessing Multi-class Pairwise Causality inference, especially when assessing medium and no causality (Table 8). Further investigation of the results shows that in these cases of misclassification, the model tends to predict *high causality* instead of *medium causality*, and *medium causality* instead of *no causality*, demonstrating that models tend to hallucinate stronger causal relationships than humans perceive. Misclassified cases can be found in Figure 6. To assess the effect of fine-tuning on models' ability to learn the Causal Discovery task, we additionally fine-tuned one encoder-only and one decoder-only language model (DeBERTa-v3-large (He et al., 2021) and Llama2-7B (Touvron et al., 2023)). The experimental setup and analysis in Appendix B show that **CRAB** also challenges the current state-of-the-art fine-tuned methods.

**Assessing Responsibility**   Going beyond pairwise causality and assessing whether LLMs can assign responsibility among potential causes of a specific event, we show that models fail to capture complex causal structures. Table 3 shows that models perform better when assessing the causality of graph structures that contain causal scores that are well-separated from each other; high causal VS low or no causal ones (*Strong Direct and Direct Causal Frames*), compared to structures with scores of various degrees (*Mixed Causal Frames*). Additionally, from Table 4, we notice a common struggle among all models regarding the Collider case; a causal diagram of two causes that contribute to an effect but are not themselves related to each other.

**Multi-document Causal Reasoning**   In Table 2, we report results for causal score prediction for both event pairs that are found in the same document and event pairs found across documents. In all experimental settings, we see better performance for event pairs extracted from the same document (in-doc pairs). Based on the in-doc high causal score percentage reported in Table 1, models tend to perform well in the causal discovery of in-doc event pairs since documents usually express causal relationships in an explicit way, likely because narrators seek to draw explicit causal links between events. Interestingly, in the MCQ setting, we find that *GPT-4* wrongly assigns the highest responsibility to events that belong in the same document

33% of the time, indicating that models themselves may be biased to prefer in-document causal links, even when humans identify a different causal link across multiple documents. This result suggests that models are able to capture causality when it is explicitly referred to in one context but struggle when it is implicitly inferred across documents.

**Memorization vs. Generalization**   Due to the lack of transparency of closed-source GPT models, concerns arise regarding whether LLMs, pre-trained on extensive internet data, were subjected to the test set of benchmarks during their pre-training phase (Jacovi et al., 2023). This presents a complex challenge in assessing the degree to which LLM performances change for events they might have seen during the pre-training phase. We study how the performance of *GPT-3/4* varies when identifying causality for real-world events occurring pre-Jan 2022 and post-Jan 2022 (the official threshold date for their training data source). Even though we provide the context to these models, we still observe a substantial drop in performance for graded causality and pairwise causality score, suggesting that the performance of models can be affected by knowledge of the events memorized during their pretraining stage.

# 7   Related Work

**Causal Reasoning Benchmarks**   There has been extensive research on introducing challenging causal benchmarks in recent years. They focus on commonsense causal discovery and reasoning (Mooij et al., 2016; Kalainathan and Goudet, 2019; Bethard et al., 2008; Dalal et al., 2023), as well as assessing causal reasoning from the perspective of plausible alternatives and counterfactuals (Roemmele et al., 2011; Frohberg and Binder, 2021; Srivastava et al., 2022; O'Neill et al., 2022). Similar to our work, there have been attempts to create benchmarks that incorporate the cross-document setup (Welbl et al., 2018; Tu et al., 2019) and causal structures (Jin et al., 2023a), but not on real-world events. Finally, domain-specific datasets that perform causal reasoning in the medical (Nori et al., 2023) and the legal (Zhong et al., 2023; Choi et al., 2023) domains have been introduced. **CRAB** differs from prior work regarding the nature of its events, real-world events, and the type of causal reasoning that assesses LLMs, actual causality.

**Assessing Causal Reasoning** Many existing studies investigate whether NLP models understand causality (Feder et al., 2022; Jin et al., 2023b; Zhang et al., 2023), providing methods to quantify the causal abilities of language models to discover causal relationships in documents (Cao et al., 2022; Yu et al., 2019; Dalal et al., 2023) and extracting cause-effect pairs as a subtask of information extraction from text (Hidey and McKeown, 2016; Huang et al., 2021; Jin et al., 2023a; Nauta et al., 2019). Another line of work studies the improvement of causal reasoning generation using instruction prompting (Kıcıman et al., 2023), Chain-of-Thought (CoT; Wei et al., 2022), and prompt augmentation (Schick et al., 2023).

## 8 Conclusions

This work introduces *CRAB*, a new causal reasoning benchmark that contains fine-grained, contextual causality annotations for ∼2.7K pairs of real-world events that describe various newsworthy stories. Using *CRAB*, we explore how LLMs assess the causal relationships between events when the causal signal comes from different contexts. Additionally, we assess LLMs performance in identifying and assessing complex causal structures. Our findings suggest that state-of-the-art language models perform poorly in pairwise causal inference and responsibility assignment when events are spread across documents. Furthermore, this weak performance is amplified when LLMs must identify causal relationships in complex causal structures rather than simple linear chains.

## 9 Limitations

As discussed in Section 3, actual causality poses a great challenge when collecting human causal judgments about real-world events. We tried to mitigate these biases by introducing intermediate validation steps throughout the data collection pipeline, keeping experts in the loop, and focusing on collecting objective causal assessments about stories grounded in the respective documents. However, several epistemic biases might remain since the news pieces that the causal judgments have been grounded on might contain biases propagated by the reporter of the story. We try to distribute this variance by using documents collected from various sources. An additional limitation of our study is related to the nature of the events of *CRAB*. Since we are collecting real-world stories that were covered by the media, it is nearly impossible to identify all possible mediating events in the causal graph. We are in a limited information situation, both from the viewpoint of the document (the document's author might be unaware of other mediating events) and of the model (we can not evaluate the amount of information stored in the weights of the model). The experiment regarding memorization and generalization is a first step toward investigating this limitation.

## Acknowledgements

We thank Negar Foroutan, Reza Banaei, Deniz Bayazit, Beatriz Borges and Mete Ismayilzada for reading and providing comments on drafts of this paper. We also gratefully acknowledge the support of the Swiss National Science Foundation (No. 215390), Innosuisse (PFFS-21-29), the EPFL Science Seed Fund, the EPFL Center for Imaging, Sony Group Corporation, and the Allen Institute for AI.

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

## A   Dataset Construction Details

### A.1   News Article Selection

Based on a selection of 20 distinctive stories, we crawl the web and select the top 20 news articles per story. When extracting articles related to a story that happened many years before, we noticed that the retrieved articles also covered recent events that were loosely related to the story's main events. Therefore, we use a time window of 9 months when extracting the articles for each story to keep only articles that have been published around the time that the respective story happened.

### A.2   Event Extraction

Using a generative approach for event extraction has two main benefits, confirmed through extensive experimental analysis. First, when prompted correctly, generative models successfully output structured information at the requested semantic abstraction, which leads to higher precision when extracting events. Second, the semantic granularity of the events we want to extract is between sentence and document level, meaning that we aim for the main events covered in the article and not syntactic events as existing works use (Ebner et al., 2020).

Similarly to Smeros et al. (2019) and Romanou et al. (2020), we filter the top 20 news articles per story. We remove the ones with less than 100 words and those with paywalls or provide re-directions to the original referred news article. Finally, we clip the article to 250 words and round to the end of the

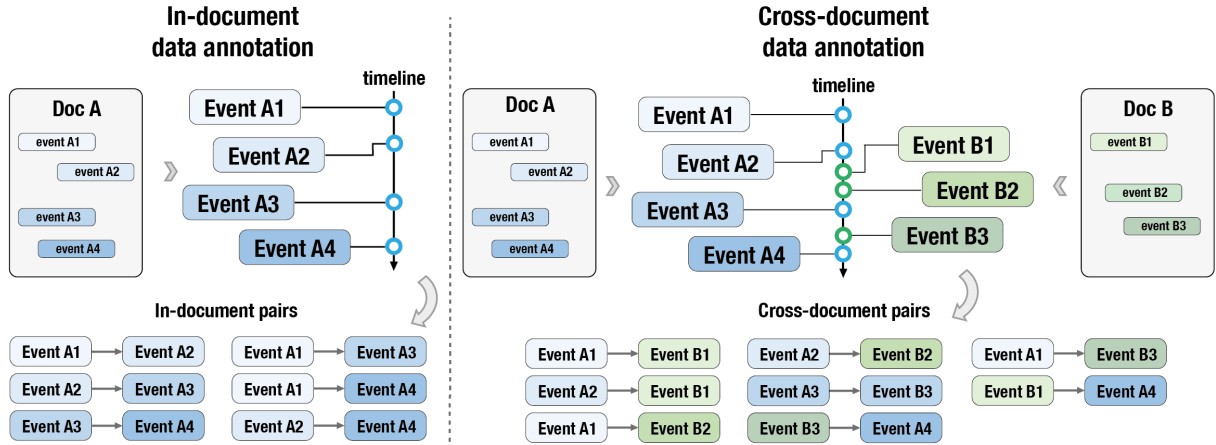

Figure 4: Dataset construction pipeline. Once events are extracted from the documents (Section 3), we order them on time and formulate story timelines. Conditioned on the temporal ordering, we create all combinations of events in the document and pass the in-document pairs to annotators. We perform a similar process for the event pairs extracted from different documents (cross-doc), including an extra step of merging document timelines into one before taking the pair combinations.

sentence token. We end up using a total number of 384 news documents as the main test bed for event extraction. Figure 17 reports statistics about **CRAB** stories.

| | |
|---|---|
| **Documents** | 173 |
| **Events** | 384 |
| **Pairs** | 2730 |
| **In-document pairs** | 360 |
| **Cross-document pairs** | 2370 |
| **Stories** | 20 |
| **Avg Timeline length** | 19 |
| **Avg pairs per story** | 137 |

Table 5: General descriptive statistics of **CRAB** regarding event and event pairs stratified for both the in-doc and cross-doc cases along with statistics about the stories and story timelines.

## A.3 Crowdsourcing Causality Scores

For the in-document annotations, we show the annotators the documents and ask them to assess 3 event pairs. For the cross-document ones, we give 2 documents at a time, along with 5 event pairs. Figure 4 depicts the pair creation process that served as input for Amazon Mechanical Turk experiments.

To select native English speakers, we focus on the group of workers whose locations are in the USA. We also ran a 2-phase qualification where we evaluated the quality of annotators on our task and selected the ones with a higher than 80% score on our qualification task. Finally, 44 out of 400 workers are selected as qualified. We pay each

worker $0.80 for doing every 3 annotations for the in-document event pairs and $0.90 for doing 5 annotations for the cross-document pairs. Figures 7 shows the screenshots of our Acceptance & Privacy Policy, and Figures 8, 9 and 10 depict the task instructions used for crowdsourcing along with the scripts for the in-doc and cross-doc tasks.

| Causal frame | # frames | # pairs | In-doc | Cross-doc |
|:---:|:---:|:---:|:---:|:---:|
| **SD** | 24 | 85 | 25 | 60 |
| **D** | 18 | 70 | 22 | 48 |
| **SC** | 30 | 254 | 26 | 228 |
| **C** | 98 | 789 | 103 | 686 |
| **NC** | 33 | 100 | 18 | 82 |
| **MC** | 149 | 1386 | 156 | 1230 |

Table 6: Number of frames for different types of causal frames (see Figure 2) along with the number of event pairs additionally stratified for the in- and cross-doc cases. SD is for Strong Direct Causality; D is for Direct Causality; SC is for Strong Contributory Causality; C is for Contributory Causality; NC is for No Causality; MC is for Mixed Causality.

| Causal chain | # Chains |
|:---:|:---:|
| **Mediation** | 774 |
| **Confounding** | 924 |
| **Collider** | 115 |

Table 7: Number of chains for different types of causal chains (see Figure 2).

| Causal Strength | Precision | Recall | F1 | Support |
|:---:|:---:|:---:|:---:|:---:|
| **High** | 0.57 | 0.61 | **0.59** | 722 |
| **Mid** | 0.35 | 0.38 | 0.36 | 685 |
| **Low** | 0.55 | 0.45 | 0.49 | 968 |
| **No** | 0.34 | 0.43 | 0.38 | 324 |

Table 8: Performance scores of *Pairwise Causality (4 classes) Inference* task for each class for the *GPT-4* model.

| Causality classes | Size | $\alpha$ |
|:---:|:---:|:---:|
| [1, 2] | 1310 | 0.04 |
| [2, 3] | 1295 | 0.08 |
| [3, 4] | 1429 | 0.12 |
| [1, 4] | 1444 | 0.38 |
| [1, 2, 3, 4] | 2730 | 0.28 |

Table 9: Krippendorff's $\alpha$ for different groups of classes.

## A.4 Inter-rater agreement

We have a total of 44 workers scoring the causality between 2730 pairs of events. All pairs are annotated by at least 7 annotators and, at most, 10 (around 21.3k annotations). We divide the causality score into 4 equal classes and compute Krippendorff's $\alpha$. We consider the ground truth as the majority vote. Table 9 shows Krippendorff's $\alpha$ for different groups of classes. As expected, the agreement to discriminate between the lowest causality class and the highest one is the highest, while it is harder for annotators to agree on discriminating between nearby classes. Krippendorff's $\alpha$ for all classes is 0.27, and 0.33 for the further classes. We note that the high number of annotators per sample increases the raw number of disagreements. Moreover, contrary to classical annotation situations where a small number of annotators label each sample, the Amazon Turk settings involve many different annotators participating in a task. Thus each sample is annotated by different workers, augmenting the variance and decreasing the agreement rate.

## B Experimental Results with Fine-Tuned LMs

We initially evaluated our proposed dataset on decoder-only models because decoder-only models (especially API-based ones such as GPT-3 / 3.5 / 4) have become important pillars of AI products, motivating researchers to benchmark their capabilities and identify their biases and limitations. However,

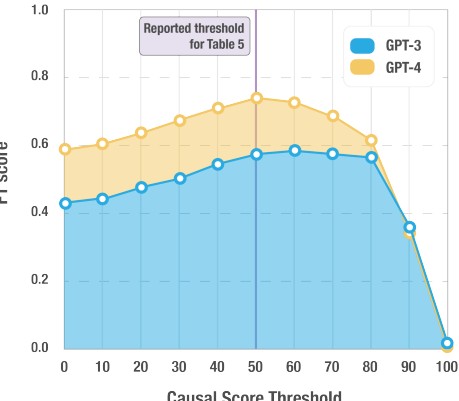

Figure 5: F1 score per threshold for the *Binary Pairwise Causality* assessment described in Section 5.2. The scores reported in Table 2 correspond to a causal score threshold of 50.

it is important to additionally evaluate our causal benchmark on different architectures and inference techniques, providing additional insight into the difficulty of the task. On that note, we fine-tuned DeBERTa-v3-large (He et al., 2021) and Llama2-7B (Touvron et al., 2023) models.

We fine-tune both models on the 3 different pairwise causality tasks presented in our paper. For each task, we create 3 different train/test splits (75%/25% ratio) to study the generalization ability of the models after fine-tuning; Date: we select 5 out of the 20 most recent stories for the test set and the rest for the train, Story: we randomly select 5 stories for the test set and the rest for the train, and Random: we randomly split the event pairs dataset, regardless of the story or the date.

The results in Table 11 show that fine-tuned DeBERTa-large (encoder-only) models fail to perform well on CRAB, showing that our benchmark challenges the current state-of-the-art fine-tuned methods. Compared to decoder-only models in a few-shots setting, DeBERTa-large tends to underperform when splitting by story, except for the easier binary pairwise causality prediction task. Additionally, as expected, the experiments with the random data split have higher scores, which validate the information leakage of the context from the train to test set and verify that models rely on the context (articles) when assessing the causal relationship of the two events. A subsequent study on how fine-tuning improves pre-trained LLMs causal reasoning abilities would be interesting, and we hope that our paper provides a strong benchmark for pursuing this research direction.

| Few-shot event extraction on | Precision | Recall | F1 |
|---|---|---|---|
| Extractive summaries with SBERT (Reimers and Gurevych, 2019) | 45.8% | 31.9% | 37.6% |
| Extractive summaries with GPT-3 (Brown et al., 2020) | 63.8% | 59.1% | 61.3% |
| Abstractive summaries with GPT-3 Brown et al. (2020) | 74.2% | 52.8% | 61.7% |
| Original Document - 5 par | **75.0%** | **65.3%** | **69.8%** |

Table 10: Zero-shot performance score for Event Extraction using the GPT-3 model. Results are based on a manually constructed dataset of 88 articles. We use different context settings by prompting the model with both providing the document in its original form as well as its summary. We found that prompting with the original text yields better quantitative and qualitative results.

| Causality Tasks | Model | Split by Date | Split by Story | Random split |
|---|---|---|---|---|
| **Pairwise Causality Score** | DeBERTa-large | 21.6 | 21.4 | 22.9 |
| | Llama2 7B | **24.3** | **26.3** | **32.8** |
| **Multi-class Pairwise Causality** | DeBERTa-large | **29.4** | **35.8** | **60.8** |
| | Llama2 7B | 23.2 | 23.1 | 32.7 |
| **Binary Pairwise Causality** | DeBERTa-large | **62.5** | **74.2** | **76.6** |
| | Llama2 7B | 51.1 | 51.9 | 58.5 |

Table 11: Macro F1-scores on test set for fine-tuned models on all Pairwise Causality Inference tasks as presented in Section 5.2. We stratify the results for Date, Story and Random splits as described in paragraph B. The best performance for each causality task is bolded.

---

**PROMPT: Event Extraction**

```
You are a journalist who whats to extract
the main events from articles.
These events need to summarize the story
of the article.
These events need to be in a list format.

<DOCUMENT>
```

Table 12: GPT-3 prompt for extracting events from documents.

---

**PROMPT: Binary Pairwise Causality**

```
You are a helpful assistant for causal
relationship understanding.
Think about the cause-and-effect relationships
related to context.

Context:
<DOCUMENTS>

Event 1: <EVENT 1>
Event 2: <EVENT 2>
Did Event 1 cause Event 2 to happen?
Please answer in a single word: yes or no.
```

Table 13: Prompt for the *Binary Pairwise Inference* task.

---

**PROMPT: Multi-class Pairwise Causality - 4 Classes**

```
You are a helpful assistant for causal
relationship understanding.
Think about the cause-and-effect relationships
related to context.

Context:
<DOCUMENTS>

Event 1: <EVENT 1>
Event 2: <EVENT 2>
How much did event 1 cause event 2 to happen?
[A] High causality: Event 1 is definitely
responsible for Event 2.
[B] Medium causality: Event 1 might have
been responsible for Event 2.
[C] Low causality: The context gives a
little indication that there is a connection
between the two events, but background info
might suggest a low causal connection.
[D] No causality: Events are somehow related
but definitely NOT causally related.

Let's work this out in a step-by-step way
to be sure that we have the right answer.
Then provide your final answer within the
tags, <Answer>A/B/C/D</Answer>.
```

Table 14: Prompt for the *Multi-class Pairwise Inference* task.

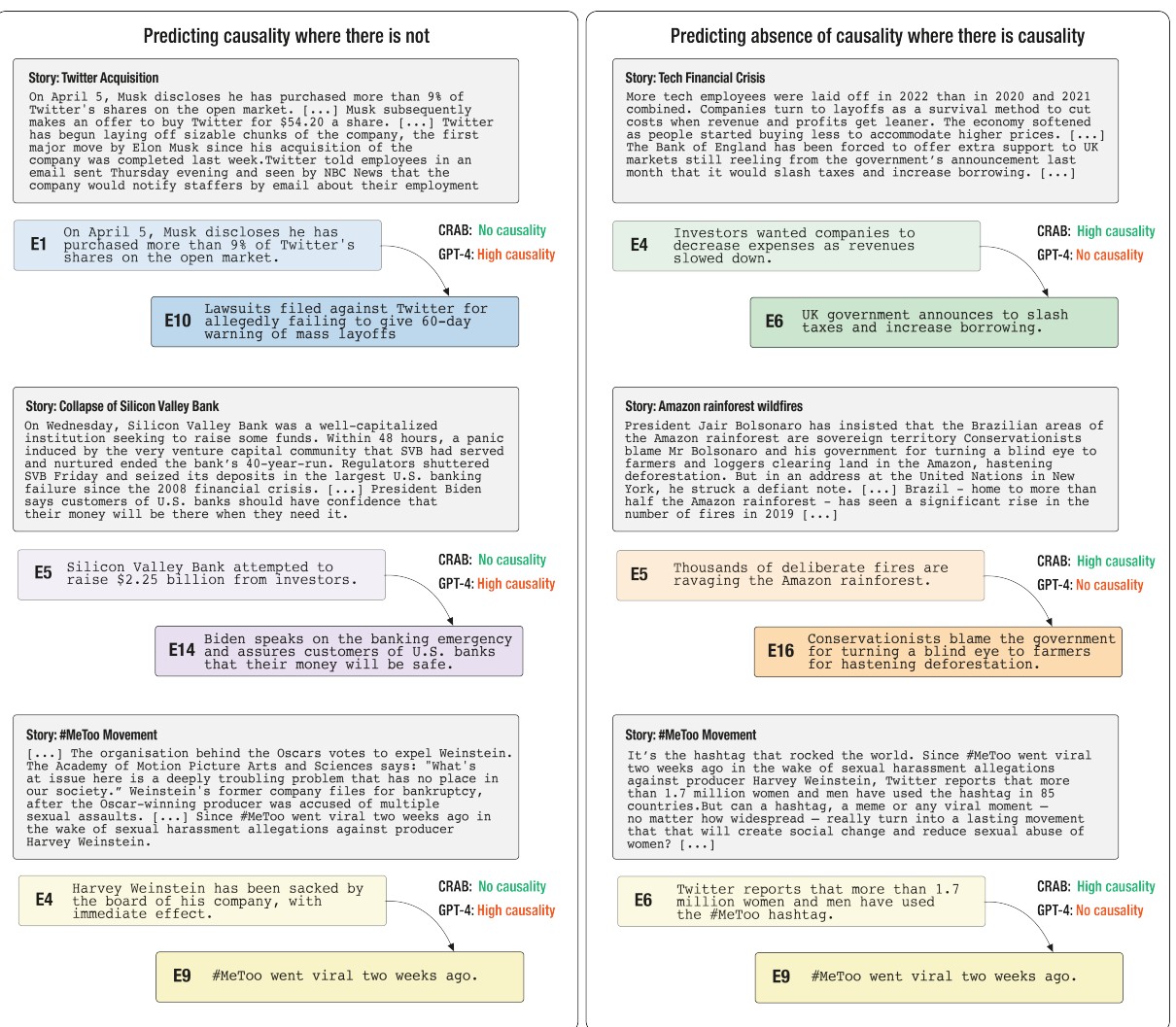

Figure 6: Failed predictions of GPT-4 regarding *CRAB* on high and no causal classes.

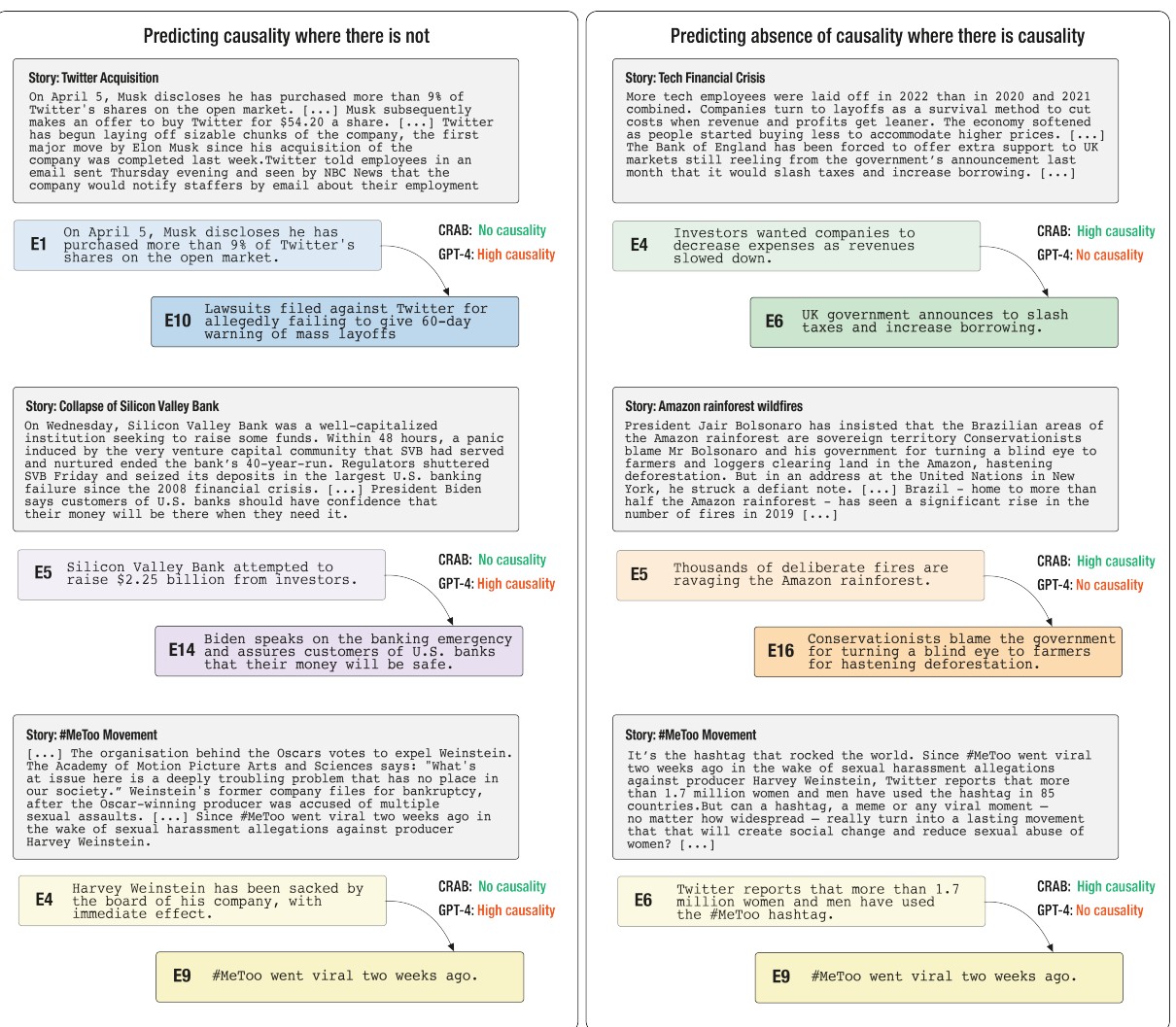

Figure 7: Acceptance and Privacy Policies script for the Amazon mTurk experiment.

Thanks for participating in this HIT!

You will be given one segment of a news article story ( news piece ) that was published in the last ten years. You will be also given **3** pairs of events described in these news piece. The goal is to determine how much one event caused the other (**causal relationship**).

You will define the **causal relationship** (how much an event causes another one) between the events by choosing a **score** from **0** (No causal relationship) to **100** (strong causal relationship) **BASED ON THE NEWS PIECE**.

The score can be interpreted as:

- **Score** 81-100 : Event A is **definitely responsible** for Event B.
- **Score** 51-80 : Event A is **responsible** for Event B, **but not directly**: there are one or more intermediate events between Event A and B that could have led to Event B happening.
- **Score** 21-50 : The context doesn't give any clue about the connection between the two events; there might be a situation where there is a **small responsibility** of Event A to cause Event B.
- **Score** 0-20 : Events are somehow related but **definitely NOT causally related.**

The **EXAMPLE** below showcases different causality scores for various event pairs.

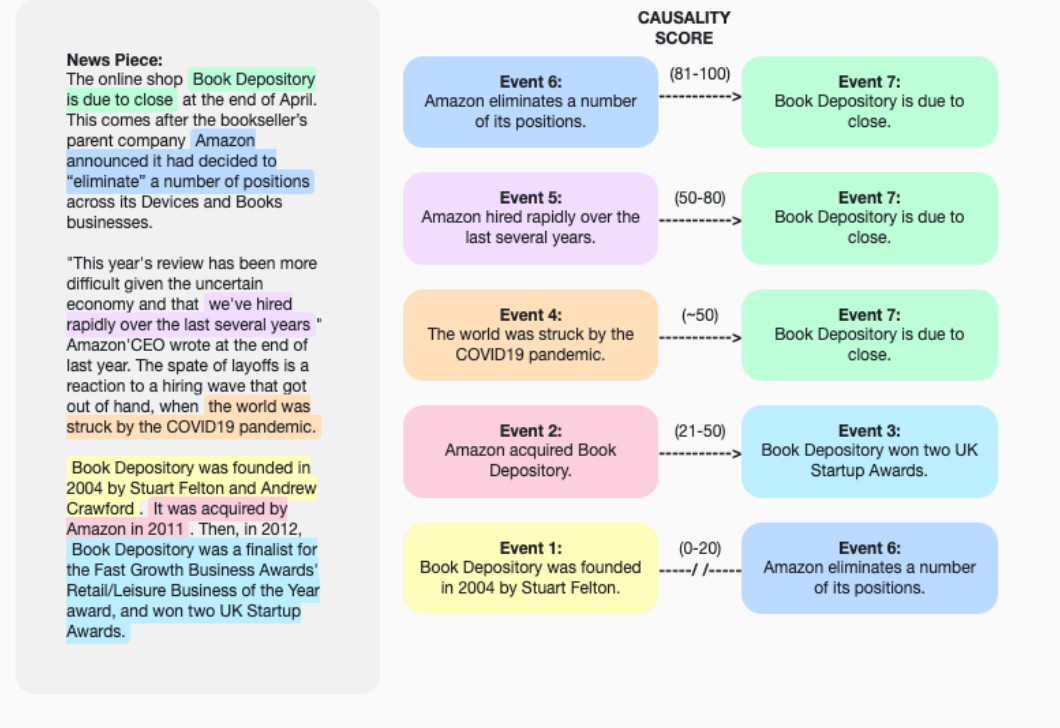

Along with the causality score, you will be asked **whether the second event would have happened if the first event didn't occur**. If your causal score in the previous question is relatively high, the second event probably couldn't have happened without the first one.

Figure 8: Amazon mTurk instructions script. Annotators need to perform the pairwise causality assessment task based on these instructions.

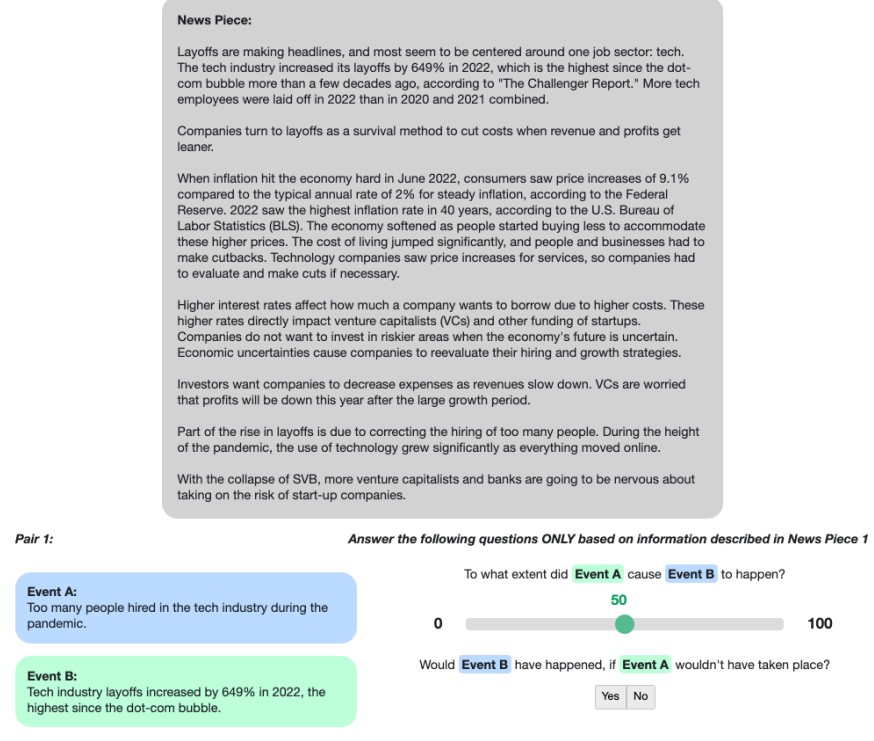

Figure 9: Amazon mTurk example script for annotating event pairs extracted from the same document. Based on the instructions, annotators need to read the narrative and assess the causal relationship of at most 3 event pairs at a time. Due to space limitations, the figure depicts only one pair.

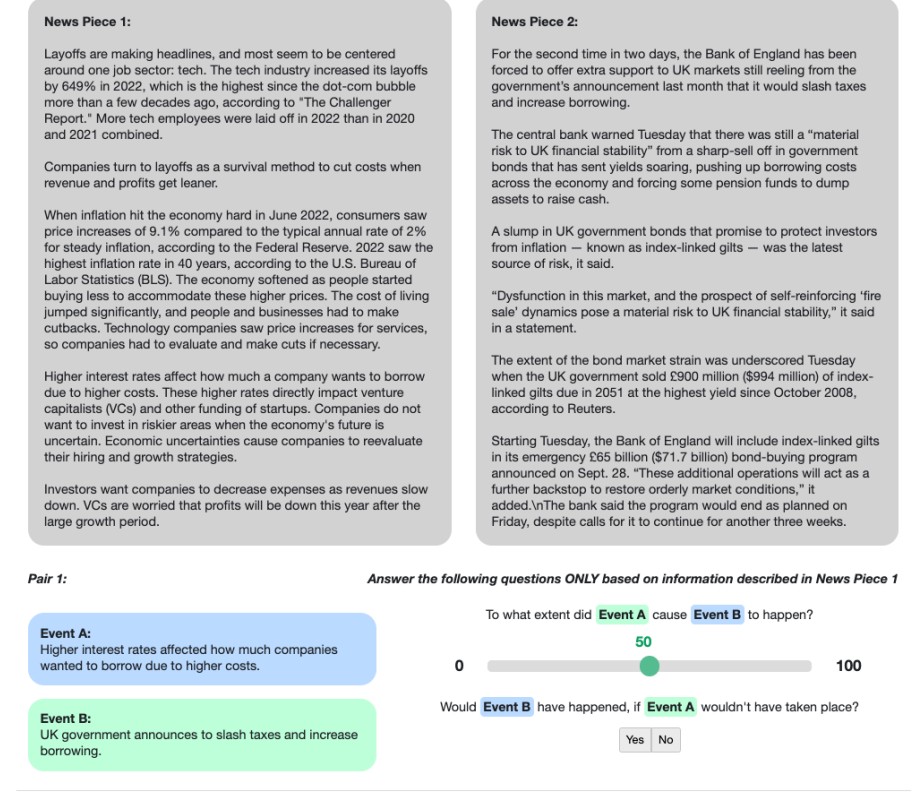

Figure 10: Amazon mTurk example script for annotating event pairs extracted from different documents. Based on the instructions, annotators need to read both narratives and assess the causal relationship of at most 5 event pairs at a time. Due to space limitations, the figure depicts only one pair.

---

**PROMPT: Graded Pairwise Causality - MCQ**

---

```
You are a helpful assistant for causal
relationship understanding.
Think about the cause-and-effect relationships
related to context.

Context:
<DOCUMENTS>

Event: <EFFECT>

What is the most likely cause of this event?
[A] <CAUSE 1>
[B] <CAUSE 2>
[C] <CAUSE 3>
[D] <CAUSE 4>

Let's work this out in a step-by-step way
to be sure that we have the right answer.
Then provide your final answer within the
tags, <Answer>A/B/C/D</Answer>.
```

---

Table 15: Prompt for the *Graded Pairwise Inference* task.

---

**PROMPT: Pairwise Causality Score**

---

```
You are a helpful assistant for causal
relationship understanding.
Think about the cause-and-effect relationships
related to context.

Context:
<DOCUMENTS>

Event 1: <EVENT 1>
Event 2: <EVENT 2>

What is the causality score between Event 1 and
Event 2 from 0 to 100?
Score above 80: Event 1 is definitely responsible
for Event 2.
Score between 50-80: Event 1 might have been
responsible for Event 2.
Score lower than 50 Events are somehow related
but definitely NOT causally related.

Let's work this out in a step-by-step way
to be sure that we have the right answer.
Then provide your final answer within the
tags, <Answer>score</Answer>.
```

---

Table 16: Prompt for the *Pairwise Causality Score Inference* task.

| Story | Number of pairs | Category |
|---|---|---|
| Twitter Acquisition | 216 | Business |
| Collapse of Silicon Valley Bank | 177 | Business;Economy |
| Gwyneth Paltrow ski crash | 174 | Lifestyle;Legal |
| Renewable energy | 66 | Environment;Economy |
| Acquisition of Credit Suisse by UBS | 147 | Business;Economy |
| Second Pink Tide | 108 | Politics |
| Release of ChatGPT | 130 | Technology;Business |
| Roe v wade Overturned | 153 | Society;Politics |
| Energy crisis | 148 | Economy |
| Heat waves | 150 | Environment |
| Tech Financial crisis | 119 | Economy;Business |
| Twitter Acquisition | 117 | Business |
| European floods | 135 | Environment |
| Ethereum Price | 119 | Business |
| Black Lives Matter riots | 133 | Society;Politics |
| Amazon rain forest wildfires | 142 | Environment;Politics |
| Notre Dame Fire | 163 | Society |
| MeToo movement | 216 | Society;Politics |
| Equifax data breach | 193 | Politics;Business |
| Panama Papers | 139 | Politics |

Table 17: Stories present in *CRAB* along with statistics about the number of pairs event each contains.