# OpenReview forum: "CRAB: Assessing the Strength of Causal Relationships Between Real-world Events"
_EMNLP/2023/Conference — EMNLP 2023 Main_

### Official Review · Reviewer_GU2u · 2023-08-04

**Soundness:** 4

**Excitement:**

3: Ambivalent: It has merits (e.g., it reports state-of-the-art results, the idea is nice), but there are key weaknesses (e.g., it describes incremental work), and it can significantly benefit from another round of revision. However, I won't object to accepting it if my co-reviewers champion it.

**Paper Topic And Main Contributions:**

The main contribution of this paper is the dataset described in it. The authors explain how they created a dataset consisting of real world events and causal relations between them. These events were extracted from web pages using a combination of automated extraction and hand-labeling, and causal relations were further annotated by annotators.

The authors also provide an analysis of the dataset created as well as simple experiments using generative models, and from the results obtained, conclude that those have difficulties recognizing actual causalities in the dataset,

**Questions For The Authors:**

Q1: What was the criteria used by the annotators to score the causal relations ?

Q2: Considering that the number of documents is not very important (173 short news articles), why go to the trouble of having experts filter events extracted by a generative model while you could have the experts read the documents and extracted the events themselves ?

Q3: why use an ill-suited generative model for evaluation while you could have trained even a simple model using the data you created ? Maybe even a CNN would have done a better job than the models you evaluated.

**Reasons To Accept:**

The principal contribution of this paper is obviously the proposed dataset. While it it of relatively small size (2700 pairs), it deals with real-world events and with causal relation expressed in separate documents, which makes it interesting for research. The annotation seems to be well done.

**Reasons To Reject:**

The way the dataset was created is limited: it consists of event causalities related to only 20 "stories" (like the buying of Twitter by Elon Musk), and all events were extracted from a set of 173 documents.

Many details are also missing about the dataset creation method: how were the stories selected ? Who chose the events after the automated generation step ? And the most important, what was the criteria for the annotators to annotate the causal relations, i.e. on what basis did they score the causal relations ? Some of these details are described in the appendix but should be present in the paper itself.

Finally, the conclusions drawn, that "most systems achieve poor performance on the task", are based on prompting existing GPT-based models to score event causalities input to them. As the models were not fine-tuned for this task, the chance that they would perform poorly under this evaluation is quite high. Why not train a simple model with the data created for this paper ?

**Reproducibility:**

3: Could reproduce the results with some difficulty. The settings of parameters are underspecified or subjectively determined; the training/evaluation data are not widely available.

**Reviewer Confidence:**

4: Quite sure. I tried to check the important points carefully. It's unlikely, though conceivable, that I missed something that should affect my ratings.

**Typos Grammar Style And Presentation Improvements:**

The following questions are not to be answered, and should serve as guide to add missing information and details to the paper.

l218: how were the stories selected ?

l223: how were the documents clipped ?

l237: higher than what ?

How were the events ordered in a timeline ?

l270: how did you end up with 2370+360=2730 pairs ? Given about 19 events per story, this means that you can make 19*18/2=171 event pairs per story, for at least 3420 pairs total.

l315,319: how were the causal frames/chains extracted and/or annotated ?

l349: what is the prompt given to the model ?

l380: what is the score-label mapping ?

l437: is that really "various" models ? The three models tested are all variations of the same (GPT).

---

> ### Author Rebuttal · Authors · 2023-08-29
>
> We appreciate the reviewer describing our proposed dataset as interesting for the research community and highlighting that our annotation process is well implemented.
>
> **Depth over breadth:** We appreciate the reviewer highlighting that the strength of the proposed benchmark is focusing on testing LLMs in various complex causality structures on real-world events expressed in separate documents. To that note, the main goal of this benchmark is to focus on depth over breadth by capturing detailed and robust annotations (7-10 annotators per pair, 21300 annotations in total, for 2730 pairs of events coming from 173 documents) with high diversity in causal structures (multi-hop causal reasoning chains, complex causal graph structures).
>
> **Additional clarifications for the dataset creation method:** The reviewer asks for further clarifications on the dataset construction. Stories were selected to cover a variety of categories like business, economy, environment, politics, etc. A detailed breakdown of the category corresponding to each story can be found in Table 16 of our paper. We manually selected top stories based on the Yearly Google Trends for the past 10 years. Due to limited resources, we selected 20 stories for annotation, aiming at a high diversity among story categories. After the automated event extraction, as described in Section 3.3 of our paper, three NLP experts on causal reasoning tasks manually filtered out events that were not valid. Then, we asked mTurk workers to read one or two articles containing the events in the causal relationship that they needed to assess. Based only on the context (articles) presented to them, they provided a score from 0 to 100 of their judgment regarding the causal strength between the events. To ensure the understanding of the tasks by the annotators, we provided a detailed explanation and breakdown of the score and various annotated examples that can be found in the Appendix [A1] of our paper. Additional details on each step of the dataset creation process can be found in our paper's Appendix [A]. As suggested by the reviewer, some of these details should be present in the main paper as well, and therefore, we will incorporate these updates for the final version.
>
> **Semi-automated event extraction:** The reviewer asks why we did not rely on humans to extract events from stories, and used GPT3 instead. We noticed that manual event extraction took significant time for each article (around 20-30 mins per article- 57 hours for 1 person). Additionally, the level of abstraction of events and their format was not uniform across different annotators. Due to the limited experts we had, we decided that filtering the automatically generated events (annotation time ~3 mins per article) was more efficient, especially upon scaling it to the full benchmark size (9 hours for 1 person vs. 57 initially).
>
> **Fine-tuning on benchmark and going beyond prompting decoder-only models.** We initially evaluated our proposed dataset on decoder-only models because decoder-only models (especially API-based ones such as GPT-3 / 3.5 / 4) have become important pillars of AI products, motivating researchers to benchmark their capabilities and identify their biases and limitations. However, as the reviewer reasonably pointed out, it is important to additionally evaluate our causal benchmark on different architectures and inference techniques, providing additional insight on the difficulty of the task.
>
> On that note, we fine-tune the following SoTA models:
>
> - DeBERTa-v3-large (encoder-only)
> - Llama2-7B (decoder-only)
>
> We fine-tune both models on the 3 different pairwise causality tasks presented in our paper. For each task, we create 3 different train/test splits (75%/25% ratio), to study the generalization ability of the models after fine-tuning:
>
> - Date: We select 5 out of the 20 most recent stories for the test set and the rest for the train.
> - Story: We randomly select 5 stories for the test set and the rest for the train.
> - Random: We randomly split the event pairs dataset, regardless of the story or the date.
>
> The F1 scores of the 18 new experiments are included in the table below:
>
> | CAUSALITY TASKS                | MODEL         | Split by date | Split by Story | Random split |
> |--------------------------------|---------------|:-------------:|:--------------:|:------------:|
> | Pairwise Causality Score          | DeBERTa-large  |      21.6      |      21.4        |     22.9      |
> |                                                   | Llama2 7B         |    **24.3**   |    **26.3**    |   **32.8**   |
> | Multi-class Pairwise Causality  | DeBERTa-large  |    **29.4**   |    **35.8**    |   **60.8**   |
> |                                                   | Llama2 7B         |      23.2       |      23.1       |     32.7       |
> | Binary Pairwise Causality         | DeBERTa-large  |    **62.5**   |    **74.2**    |   **76.6**   |
> |                                                   | Llama2 7B         |      51.1       |      51.9       |     58.5       |
>
> The results show that fine-tuned DeBERTa-large (encoder-only) models fail to perform well on CRAB, showing that our benchmark challenges the current SoTA methods. Compared to decoder-only models in a few-shots setting, DeBERTa-large tends to underperform when splitting by story, except for the easier binary pairwise causality prediction task. Additionally, as expected, the experiments with the random data split have higher scores, which validate the information leakage of the context from the train to test set and verify that models rely on the context (articles) when assessing the causal relationship of the two events. A subsequent study on how fine-tuning improves pre-trained LLMs causal reasoning abilities would be interesting, and we hope that our paper provides a strong and necessary benchmark for pursuing this research direction.
>
>
> **References:**
>
> Chung, Hyung Won, et al. "Scaling instruction-finetuned language models." arXiv preprint arXiv:2210.11416 (2022).

---

### Official Review · Reviewer_g6sm · 2023-08-04

**Soundness:** 4

**Excitement:**

3: Ambivalent: It has merits (e.g., it reports state-of-the-art results, the idea is nice), but there are key weaknesses (e.g., it describes incremental work), and it can significantly benefit from another round of revision. However, I won't object to accepting it if my co-reviewers champion it.

**Paper Topic And Main Contributions:**

This paper introduces CRAB, so named after 'Causal Reasoning Assessment Benchmark', that has been devised to track causal understanding of events in narratives. CRAB contains about 2.7k event pairs
CRAB collects causal subgraphs, where events are linked with their potential causes, and labeled (by humans) with causal scores. The paper proposes another view on events and their causes (causal chains), that is chains of temporarily ordered events that led to a further event. 20 stories about major recent events were selected, and news articles collected; the events underwent a pipeline whereby they were extracted, verified, timelines were identified and event disambiguated. Finally, causal links between all (2730) event pairs were individuated, with involved events belonging to either same document or different documents.

All models perform poorly on CRAB, with GPT-4 showing a higher performance in most tasks compared to GPT-3 and Flan-Alpaca. However, the Authors admit that models perform worse on causal reasoning when events are derived from complex causal structures compared to simple linear causal chains, and that overall models fail to capture complex causal structures.

The paper is overall clear and I really enjoyed reading it. However, I feel that Authors should discuss one main issue. LM are basically probability distributions over text sequences: as such, LMs are unfit to deal with causal reasoning or matters such as acquiring any sort of propositional knowledge (what is needed to determine whether an event did influence another one).
It is not surprising that the models at stake obtained poor performance in the evaluation. Authors should then explain how a LM can be expected to track causal relations. This point should be discussed in the Introduction and/or discussed after having presented the experimental results.



Questions & Concerns

- Question on Figure 1. Based on this Figure, one realizes that E1 (that is Musk offered to by Twitter for dollars 44 billion in April) causes in 'low causality' fashion E2 (that is 'Elon Musk closes dollars 44 billion deal to buy Twitter'). Provided that I am not an expert of causal reasoning, this does not seem a real cause. Maybe having a sound notion of what is causal and what is not may be helpful in selecting the appropriate examples, and spread beneficial effects on resources and on LMs training steps.
- Section 3.5 Causality Strength Validation. "we perform a validation of the collected causal scores by asking three expert annotators to further annotate event pairs with high variance". Does this mean that final figures were obtained based only on expert annotators ratings? or how were ratings merged?
- How many events are featured by high variance?
- Figure 2 is referred many times, but there is no Figure 2. At times, Figure 3 seems compatible with information referred to as available in Figure 2, but not always. Please correct this mistake.
- lines 454:458. "Both models tend to predict causation between events with a gold causal score  above 50. Interestingly, if we increase this gold  score threshold and consider only causal events with a high causal connection, the performance in all models drops". How do the Authors interpret this trend?


**Questions For The Authors:**

[Q1] Question on Figure 1. Based on this Figure, one realizes that E1 (that is Musk offered to by Twitter for dollars 44 billion in April) causes in 'low causality' fashion E2 (that is 'Elon Musk closes dollars 44 billion deal to buy Twitter'). Provided that I am not an expert of causal reasoning, this does not seem a real cause. Maybe having a sound notion of what is causal and what is not may be helpful in selecting the appropriate examples, and spread beneficial effects on resources and on LMs training steps.
[Q2] Section 3.5 Causality Strength Validation. "we perform a validation of the collected causal scores by asking three expert annotators to further annotate event pairs with high variance". Does this mean that final figures were obtained based only on expert annotators ratings? or how were ratings merged?
[Q2.2] How many events are featured by high variance?
[Q3] Figure 2 is referred many times, but there is no Figure 2. At times, Figure 3 seems compatible with information referred to as available in Figure 2, but not always. Please correct this mistake.
[Q4] lines 454:458. "Both models tend to predict causation between events with a gold causal score  above 50. Interestingly, if we increase this gold  score threshold and consider only causal events with a high causal connection, the performance in all models drops". How do the Authors interpret this trend?


**Reasons To Accept:**

- the paper proposes an interesting and useful resource, which seems to be challenging for NLP systems.


**Reasons To Reject:**

- The approaches employed to deal with the tracking of causal relations seem to be far from reaching satisfactory results; Authors were solicited to elaborate on how they expect that LMs can be employed to track causal relations.

**Reproducibility:**

4: Could mostly reproduce the results, but there may be some variation because of sample variance or minor variations in their interpretation of the protocol or method.

**Reviewer Confidence:**

4: Quite sure. I tried to check the important points carefully. It's unlikely, though conceivable, that I missed something that should affect my ratings.

---

> ### Author Rebuttal · Authors · 2023-08-29
>
> We thank the reviewer for acknowledging our proposed benchmark as a challenging and interesting resource.
>
> **How causal knowledge can be integrated into LLMs?**: Our proposed CRAB can be viewed as a causal graph that can be used by LLMs either as external knowledge (Zhang et al., Bosselut et al.) or as structured information incorporated in the model’s prompt (Yao et al., Schick et al.) as previous literature suggests. We look forward to future works on how they can integrate neural-symbolic methods that perhaps improve the causal reasoning performance on our benchmark. We thank the reviewer for suggesting further elaboration on the future work, and we will update the final version of the paper accordingly.
>
> **Details on Figure 1**: The reviewer discussed the relationship between E1 (“Musk offered to buy Twitter for 44 billion dollars in April”) and E2 (“Elon Musk closes 44 billion dollars deal to buy Twitter.”). We clarify that Figure 1 should be read as all events E1-5 leading to E6. The causality levels mentioned in the boxes are respective to the events only towards E6: E1 and E3 have low causality on E6; E2 has medium causality on E6;  E4 and E5 have high causality on E6.  The additional arrows represent temporal dependency (not necessarily causality) between events. We thank the reviewer for noticing that a better presentation could improve the message of the figure, and we will update it accordingly.
>
> **Expert validation process**: The reviewer asks for more details on the process we followed for validating mTurk annotations by experts. After crowdsourcing the causal scores from Mechanical Turk, we selected scores that fell on the boundary of different causal classes (high, medium, low, no causality) and also had high variance. This subset of event pairs consists of ~26.7% of the total benchmark. The experts were asked to validate these high-variance edge cases by choosing which of the neighboring classes was a better class for the annotations. We will update the final version of the paper by including this information in the main section of the benchmark creation.
>
> **Figure 2 can be found at the top of page 2.** We assume that maybe there was a rendering issue for that figure. We will try to optimize figure sizes and ensure proper image rendering for the final version of the paper.
>
> **Interpretation of causality threshold on models’ performance**: The reviewer suggested that further justification should be added regarding the performance curve of the models when the binary causality threshold changes. The binary causality threshold is the threshold we put on the causality score (0-100) to consider it causal. The more we increase the score threshold, the more precision drops (as the model’s prediction threshold remains the same), and the overall models’ performance (F1 score) drops. We thank the reviewer for this comment, and we will additionally include the precision and recall curves in the final version of the paper. A possible interpretation behind these results is that large language models are typically quite good at identifying the distributional similarity between concepts. When it comes to making causal judgments, they may identify multiple related events as causally associated with an outcome rather than a single consequential event. This means that they may misclassify events weakly related to the event in question as causes. Ideally, when asked about a binarized causal relationship, we would like models to provide only the main causes (strong relation between two events).
>
> **References**:
>
> Zhang, Xikun, et al. "Greaselm: Graph reasoning enhanced language models for question answering." arXiv preprint arXiv:2201.08860 (2022).
>
> Yao, Shunyu, et al. "React: Synergizing reasoning and acting in language models." arXiv preprint arXiv:2210.03629 (2022).
>
> Schick, Timo, et al. "Toolformer: Language models can teach themselves to use tools." arXiv preprint arXiv:2302.04761 (2023).
>
> Bosselut, Antoine, et al. "Dynamic neuro-symbolic knowledge graph construction for zero-shot commonsense question answering." Proceedings of the AAAI conference on Artificial Intelligence. Vol. 35. No. 6. (2021).

---

### Official Review · Reviewer_d7EN · 2023-08-05

**Soundness:** 4

**Excitement:**

4: Strong: This paper deepens the understanding of some phenomenon or lowers the barriers to an existing research direction.

**Paper Topic And Main Contributions:**

This paper introduces a new dataset “Causal Reasoning Assessment Benchmark” which consists of 2,730 pairs of real-world events. The performances of GPT models on the new dataset suggest that these causality tasks are difficult for even state-of-the-art large language models.

**Questions For The Authors:**

1. What were the qualification task or acceptance criteria for expert annotators mentioned on the paper? What are their annotator agreement values? How are they different from Amazon mTurkers in terms of qualification?
2. Why did not use encoder models? Causality predictions can be mapped as a QA model task?
3. Why didn't you try fine-tuning the models? This can provide more insights about the difficulty of the task

**Reasons To Accept:**

- The paper is well-written and their labeling methods and data statistics are thoroughly explained and analyzed.
- They are releasing a new dataset.
- Labeling tasks, qualification tasks, GPT prompts and other related details are explained well.


**Reasons To Reject:**

Annotation agreement values are low (although causality level scoring is inherently a hard task as mentioned in the paper, annotation guidelines or task design could be improved to increase the agreement).



**Reproducibility:**

5: Could easily reproduce the results.

**Reviewer Confidence:**

3: Pretty sure, but there's a chance I missed something. Although I have a good feel for this area in general, I did not carefully check the paper's details, e.g., the math, experimental design, or novelty.

**Typos Grammar Style And Presentation Improvements:**

- Figure 2: The top left side first two figures ‘Dirent’ -> ‘Direct’.
- L869: Missing a closing parenthesis.
- Table 11: You are a journalist who ‘whats’ -> ‘wants’

---

> ### Author Rebuttal · Authors · 2023-08-29
>
> We thank the reviewer for their comments, and for describing our benchmark creation process as thoroughly explained and analyzed. Below, we answer concerns and questions regarding the annotator agreement score, the expert annotation qualification and agreement, and the fine-tuning of encoder models to tackle the tasks.
>
> **Annotation agreement scores**: The reviewer points out the limitation of mTurk annotators agreement scores. To tackle this in our work, (1) not only did we use 7 to 10 annotators per event pair to make up for the inherent difficulty and subjectivity of the task, but also (2) we strengthened it with further expert annotation. Moreover, the low inter-rater agreement has to be taken with a pinch of salt, as the agreement score is usually computed for a unique set of annotators labeling the same samples, while with MTurk the samples are not necessarily labeled with the same set of annotators.
> From the beginning of this study, countering the subjectivity and complexity of the task to obtain robust causal scores from human annotators was a challenge at the heart of the research question. That is why we dedicated a considerable amount of time to carefully designing the task and the annotation guidelines, comparing several task designs (binary, multi-class, numerical causal score) and instructions before settling for the ones presented in the paper.
>
> **Expert qualification and agreement scores**:
> Expert annotators are NLP researchers who are familiar with the task of causal inference.  After the first round of annotations, we select event pairs for the second round according to two criteria:
> 1) High variance in causal score between mTurk annotators.
> 2) The average causal score falls close to a class threshold (from 0 to 100, the class thresholds are 20, 50, and 80; we provide the definitions of the classes to the annotators as guidance in the instruction).
>
> According to these criteria, 26.7% of the event pairs were re-annotated by experts. The inter-rater agreement, using Krippendorph’s alpha, is 0.70.
>
> **Fine-tuning encoder models**:
> The reviewer mentions that trained methods, and more specifically encoder models, might have been more competitive than generative models for this task.  We provide 6 additional experimental settings and their results on fine-tuned encoder-only and decoder-only models. More specifically, we fine-tune the following SoTA models:
> - DeBERTa-v3-large (encoder-only)
> - Llama2-7B (decoder-only)
>
> We fine-tune both models on the 3 different pairwise causality tasks presented in our paper. For each task, we create 3 different train/test splits (75%/25% ratio) to study the generalization ability of the models after fine-tuning:
>
> - Date: We select 5 out of the 20 most recent stories for the test set and the rest for the train.
> - Story: We randomly select 5 stories for the test set and the rest for the train.
> - Random: We randomly split the event pairs dataset, regardless of the story or the date.
>
> The F1 scores of the 18 new experiments are included in the table below:
>
> | CAUSALITY TASKS                | MODEL         | Split by date | Split by Story | Random split |
> |--------------------------------|---------------|:-------------:|:--------------:|:------------:|
> | Pairwise Causality Score          | DeBERTa-large  |      21.6      |      21.4        |     22.9      |
> |                                                   | Llama2 7B         |    **24.3**   |    **26.3**    |   **32.8**   |
> | Multi-class Pairwise Causality  | DeBERTa-large  |    **29.4**   |    **35.8**    |   **60.8**   |
> |                                                   | Llama2 7B         |      23.2       |      23.1       |     32.7       |
> | Binary Pairwise Causality         | DeBERTa-large  |    **62.5**   |    **74.2**    |   **76.6**   |
> |                                                   | Llama2 7B         |      51.1       |      51.9       |     58.5       |
>
> The results show that fine-tuned DeBERTa-large (encoder-only) models fail to perform well on CRAB, showing that our benchmark challenges the current SoTA methods. Compared to decoder-only models in a few-shots setting, DeBERTa-large tends to underperform when splitting by story, except for the easier binary pairwise causality prediction task. Additionally, as expected, the experiments with the random data split have higher scores, which validate the information leakage of the context from the train to test set and verify that models rely on the context (articles) when assessing the causal relationship of the two events. A subsequent study on how fine-tuning improves pre-trained LLMs causal reasoning abilities would be interesting, and we hope that our paper provides a strong and necessary benchmark for pursuing this research direction.

---

### Meta-Review · Area_Chair_V42b · 2023-09-12

**Recommendation:** 4

**Metareview:**

The paper introduces a new benchmark "CRAB" -- Causal Reasoning Assessment Benchmark, which is used to evaluate causal understanding of events in real-world narratives. The dataset contains 2,730 pairs of real-world events, which are based on a set of 20 stories (while it seems small, the annotations are non-trivial and 7-10 annotators have annotated each instance). The performances of GPT models on this dataset suggest that these causality tasks are not very good, highlighting that this is an open task.
The paper also contains analyses to provide insights into what makes the task difficult.

---

### Decision · Program_Chairs · 2023-10-07

**Decision:**

Accept-Main

**Comment:**

The paper introduces a new benchmark "CRAB" -- Causal Reasoning Assessment Benchmark, which is used to evaluate causal understanding of events in real-world narratives. The dataset contains 2,730 pairs of real-world events, which are based on a set of 20 stories (while it seems small, the annotations are non-trivial and 7-10 annotators have annotated each instance). The performances of GPT models on this dataset suggest that these causality tasks are not very good, highlighting that this is an open task.
The paper also contains analyses to provide insights into what makes the task difficult.